# The Antibiotic-Resistant Protein MfpA Modulates Host Cell Apoptosis and Promotes Mycobacterial Survival by Targeting Mitochondria and Regulating the NF-κB Signaling Pathway

**DOI:** 10.3390/cells14120867

**Published:** 2025-06-09

**Authors:** Weishan Zhang, Zheng Jiang, Kaixia Mi

**Affiliations:** 1CAS Key Laboratory of Pathogen Microbiology and Immunology, Institute of Microbiology, Chinese Academy of Sciences, Beijing 100101, China; zhangws@im.ac.cn (W.Z.); jiangz@im.ac.cn (Z.J.); 2Medical School, University of Chinese Academy of Sciences, Beijing 101408, China

**Keywords:** *Mycobacterium tuberculosis*, *Mycobacterium bovis Bacillus Calmette-Guérin*, antibiotic-resistant proteins, MfpA, mitochondria, apoptosis

## Abstract

*Mycobacterium tuberculosis* (Mtb) is a major global health threat, exacerbated by the emergence of antibiotic-resistant strains. This study investigated fluoroquinolone resistance protein A (MfpA), which enhances mycobacterial survival by targeting host mitochondria and regulating apoptosis. Wild-type (WT) and knockout (KO) *Mycobacterium bovis Bacillus Calmette-Guérin* (BCG) strains, a common model for Mtb, were utilized to examine host cell responses. Compared to WT strains, KO strains showed reduced colony-forming units (CFUs), elevated TNF-α and IL-6 levels, and increased apoptosis. MfpA was found to localize to mitochondria, increasing ROS production and disrupting mitochondrial membrane potential. Transcriptomic analysis revealed that MfpA modulated the NF-κB signaling pathway, regulating the expression of *gadd45β*. These results suggest that MfpA drives both antibiotic resistance and virulence by suppressing apoptosis via the mitochondrial and NF-κB pathways, promoting mycobacterial persistence. Studies using BCG provide valuable insight into Mtb’s survival mechanisms, highlighting MfpA’s dual role in resistance and pathogenesis.

## 1. Introduction

*Mycobacterium tuberculosis* (Mtb) is one of the leading causes of death worldwide [1], exacerbated by the emergence of antibiotic-resistant strains, which significantly complicate treatment efforts. Moreover, the global COVID-19 pandemic has contributed to an increase in drug-resistant tuberculosis (DR-TB) strains [2], thereby making TB eradication more challenging. Traditional antibiotic development has largely focused on directly inhibiting bacterial growth or killing the pathogen by targeting essential processes like cell wall synthesis (e.g., isoniazid), nucleic acid replication (e.g., fluoroquinolones), or metabolic pathways (e.g., rifampicin). While these drugs show effectiveness in initial treatments, Mtb rapidly develops resistance through mechanisms such as gene mutations, drug efflux pump activation, or modifications to drug targets. Additionally, drug-sensitive strains can survive antibiotic pressure in a dormant state (persister cells) or through phenotypic tolerance, which can eventually lead to fully resistant strains [3,4]. Unfortunately, the development of new antibiotic classes has not kept pace with the growing need for effective treatments.

A promising alternative to traditional antibiotics is to target functions critical to Mtb’s ability to cause disease, such as virulence factors [5]. Mtb has evolved complex mechanisms to evade the host’s immune system, making it a highly successful pathogen [6]. Upon infection, Mtb is primarily detected by macrophages, which engulf the bacteria through phagocytosis. Following this, alveolar macrophages may undergo apoptosis, a form of programmed cell death triggered by the activation of caspase protease systems. This mechanism helps eliminate the pathogen early, presenting apoptotic bodies to other phagocytes to aid in containing Mtb growth and spread [7]. However, Mtb has evolved multiple strategies to manipulate the host cell machinery to improve its chances of survival. The bacterium resides within the phagosome, from which secreted proteins leak through pores and target various host organelles, including mitochondria. Recent studies have demonstrated that various Mtb proteins, such as Rv1813c and Rv0674, target the host mitochondria [8,9]. These proteins modulate mitochondrial signaling, affecting key processes like apoptosis and reactive oxygen species (ROS) production, which allows Mtb to control the host cell and enhance its chances of survival within the host [10].

In addition to these survival strategies, antibiotic-resistant proteins (ARPs) in Mtb not only contribute to drug resistance but may also regulate virulence and modulate host–pathogen interactions. For example, resistance-related genes such as *katG* and *rpoB* have been shown to modulate Mtb’s virulence by interfering with the host immune response or inhibiting apoptosis, thereby improving bacterial survival within host cells. This dual function of antibiotic targets suggests that these proteins not only serve as critical sites for drug action, but also act as molecular switches controlling virulence. KatG, the target of isoniazid, mediates drug activation while simultaneously protecting Mtb from oxidative damage by ROS produced by the host [11]. Therefore, strategies focusing solely on drug resistance mechanisms might overlook the broader roles these targets play in pathogenesis, potentially selecting for more resistant and virulent strains under drug pressure.

Understanding how Mtb uses its resistance-related proteins to modulate host responses unlocks new therapeutic avenues. Targeting both resistance mechanisms and virulence pathways could help mitigate the risks of drug resistance. Combination therapies that inhibit virulence factors (like KatG) to enhance host immune responses while using antibiotics to eradicate residual bacteria [12] could represent a promising approach. However, much remains unknown about the virulence functions of many resistance-related genes, limiting our ability to fully comprehend Mtb’s adaptive survival strategies and their implications for treatment development.

Fluoroquinolones (FQs) are essential antibiotics for treating drug-resistant tuberculosis (DR-TB), targeting DNA gyrase and topoisomerase IV, which are crucial for bacterial DNA replication [13,14]. FQs cause double-strand breaks and replication fork arrest, leading to cell death [15,16]. However, mutations in these enzymes, particularly in DNA gyrase, lead to FQ resistance and increase the risk of multidrug-resistant tuberculosis (MDR-TB) [13]. A key factor in Mtb’s FQ resistance is the mycobacterial fluoroquinolone resistance protein A (MfpA), which inhibits DNA gyrase’s ATP-dependent activity, preventing FQ-induced DNA damage. While MfpA’s role in resistance has been well studied, its impact on host immune responses remains unclear [17].

This study aimed to address this gap in our understanding by investigating the dual role of MfpA in both the antibiotic resistance of Mtb and its interaction with host cells. Using *Mycobacterium bovis Bacillus Calmette-Guérin* (*M. bovis* BCG) as a model organism, we explored the regulatory effects of MfpA on host cell apoptosis and immune signaling pathways. Upon comparing the survival of wild-type (WT) and *mfpA*-knockout (KO) *M. bovis* BCG strains within infected host cells, we showed that MfpA functions as a virulent factor, affecting *M. bovis* BCG survival within the host and causing dysregulated pro-inflammatory cytokine production. MfpA was found to be secreted by the mycobacterium within infected host cells and localized to their mitochondria, where it influences reactive oxygen species (ROS) levels and mitochondrial membrane potential. Additionally, we showed that MfpA is involved in mitochondria-related apoptosis, which plays a crucial role in regulating the host immune response to Mtb infection and significantly affects bacterial survival and persistence. We found that by disrupting mitochondrial membrane potential and modulating ROS production, MfpA affects host cell apoptosis. Further transcriptomic analysis revealed that MfpA promotes Mtb survival by modulating the NF-κB signaling pathway and regulating the expression of Growth arrest and DNA damage-inducible 45 (GADD45B), a critical stress sensor involved in inflammation. These findings highlight the importance of MfpA not only in FQ resistance, but also in regulating host cell responses, providing new insights into Mtb’s ability to persist and evade host immunity by influencing apoptotic pathways.

## 2. Materials and Methods

### 2.1. Bacterial Strains and Culture Conditions

In this study, *Mycobacterium bovis* BCG Pasteur was used to investigate the functional roles of MfpA, a protein that shares 100% genetic identity with Mycobacterium tuberculosis strain H37Rv. The mycobacterial strains were grown in 7H9 medium, which consisted of Middlebrook 7H9 medium (Becton Dickinson, Sparks, MD, USA) supplemented with 10% ADS. The ADS mixture contained 5% (*w*/*v*) bovine serum albumin fraction V (Amresco, 0332-100G, Solon, OH, USA ), 2% (*w*/*v*) D-dextrose (Sinopharm Chemical Reagent, 6300518, Shanghai, China), and 8.1% (*w*/*v*) NaCl (Sinopharm Chemical Reagent, 10019318, Shanghai, China), as well as 0.5% (*v*/*v*) glycerol (Innochem, A13031-500ML, Beijing, China) and 0.05% (*v*/*v*) TWEEN 80 (Sinopharm Chemical Reagent, 30189828, Shanghai, China). The *mfpA*-knockout mutant strain was maintained in medium containing 50 mg/L hygromycin B (Roche, Indianapolis, IN, USA). Additionally, recombinant strains such as pMV361-*mfpA*/BCG::Δ*mfpA*, pMV261-*mfpA*::BCG, and pMV261-flag-*mfpA*::BCG were cultured in medium supplemented with 25 mg/L kanamycin (Amresco, OH, USA).

### 2.2. Construction of BCG::ΔmfpA, pMV361-mfpA/BCG::ΔmfpA, pMV261-mfpA::BCG, and pMV261-flag-mfpA::BCG

Specialized transduction based on mycobacteriophages was utilized to replace the *mfpA* gene and were amplified from *M. bovis* BCG genomic DNA using the primer pairs MfpA-LL/MfpA-LR and MfpA-RL/MfpA-RR. These primers are listed in Appendix A. The amplified regions were cloned into the plasmid p0004s [18], which was previously digested with *Van91*I (Thermo Fisher Scientific, Waltham, MA, USA). The resulting plasmid p0004s-*mfpA* was then linearized with *Pac*I (Thermo Fisher Scientific, Waltham, MA, USA) and inserted into *Pac*I-digested phAE159. The shuttle plasmid was transformed into *E. coli* HB101 Electro-Cells (Takara Bio Inc., 9021, Shiga, Japan) through phage packaging using MaxPlax Lambda packaging extract (Epicentre Biotechnologies, MP5120, Madison, WI, USA) [19]. The plasmids were electroporated into *Mycobacterium smegmatis* mc^2^155 for phage propagation. The correctness of the transformants were verified by PCR using MfpA-InL/MfpA-InR primers (Appendix A).

To construct complemented strains, the full-length *mfpA* gene was amplified from *M. bovis* BCG genomic DNA using the primer pair 361-MfpA-F/361-MfpA-R (Appendix A), and the PCR product was cloned into the integrating vector pMV361. The plasmid was electroporated into the ∆*mfpA*-knockout strain, resulting in the construction of pMV361-*mfpA*/BCG::Δ*mfpA*. A similar method was used to generate pMV261-*mfpA*::BCG. The pMV261-*flag-mfpA* vector was constructed by amplifying the *mfpA* gene with primers 261-FLAG-MfpA-F and 261-FLAG-MfpA-R (Appendix A) and inserting it into the multiple cloning site (MCS) of the pMV261 backbone. The Flag-tag sequence ‘DYKDDDDK’ was fused to the N-terminus of MfpA, and the plasmid was transformed into *M. bovis* BCG to generate the pMV261-*flag*-*mfpA*::BCG.

### 2.3. In Vitro Growth Kinetics of Recombinant M. bovis BCG Strains

The growth rates of BCG::Δ*mfpA*, pMV361-*mfpA*/BCG::Δ*mfpA*, pMV261-*mfpA*::BCG, and their parental strains were compared in 7H9 medium supplemented with 10% ADS. The optical density at 600 nm (OD_600_) was measured every 24 h for each strain. The experiment was repeated three times.

### 2.4. Cell Culture

Raw264.7 cells were cultured in an incubator at 37 °C with 5% CO_2_ in Dulbecco’s Modified Eagle Medium (DMEM) (Meilunbio, Dalian, China) supplemented with 10% Fetal Bovine Serum (FBS) (BOBIO, Shanghai, China), 1% penicillin–streptomycin (Biosharp, Hefei, China).

### 2.5. Intracellular Survival of Recombinant M. bovis BCG Strains

Raw264.7 cells were seeded in 24-well plates at 6 × 10^5^ cells/well (Nest Biotechnology, Wuxi, China) and cultured overnight in DMEM (Meilunbio, Dalian, China) containing 10% FBS (BOBIO, F800-050, China) at 37 °C and 5% CO_2_ to facilitate cell adherence. The cells were then infected with wild-type BCG, BCG::Δ*mfpA*, pMV361-*mfpA*/BCG::Δ*mfpA*, and pMV261-*mfpA*::BCG at a multiplicity of infection (MOI) of 5. After 4 h, the bacteria were withdrawn and extracellular bacteria were killed by adding 200 μg/mL gentamicin (Sigma Aldrich, St. Louis, MO, USA). Fresh DMEM containing 10% FBS was added, and the incubation continued for an additional 48 h. After removing extracellular bacteria with 1 × PBS (Meilunbio, Dalian, China), the cells were lysed with pre-cooled 1 × PBST: 1 × PBS containing 0.05% (*v*/*v*) Tween 80 (Sinopharm Chemical Reagent, Shanghai, China). Cell lysates were serially diluted and plated onto 7H10 (Becton Dickinson, Sparks, MD, USA) agar plates. The plates were incubated at 37 °C for approximately 3 weeks, and colony-forming units (CFUs) were counted for each infected group. The experiment was repeated three times.

### 2.6. Identification of MfpA Secretory Protein

After selecting and verifying positive clones of pMV261-*flag*-*mfpA*::BCG, the bacteria were inoculated into fresh 7H9 complete medium containing kanamycin (25 mg/L) at a 1:100 dilution. The cultures were incubated at 37 °C with shaking at 120 rpm until the OD_600_ reached 1.0. Both bacterial cells and the supernatant were harvested, and proteins were precipitated using acetone precipitation. The expression of the MfpA protein in the bacterial cells and culture supernatant was analyzed by western-blotting using an anti-Flag antibody (1:2500, Abmart, Shanghai, China).

### 2.7. Analysis of Subcellular Localization of MfpA in Eukaryotic Cells

HEK-293T cells were cultured in DMEM supplemented with 10% FBS at 37 °C with 5% CO_2_. For transfection, the cells were seeded into glass-bottomed culture dishes (Nest Biotechnology, Wuxi, China) at a density of 5 × 10^5^ cells per well and incubated for 24 h to reach 70–80% confluency. The eukaryotic expression vector pcDNA3.1(+)-*egfp*-*mfpA* was diluted in Opti-MEM^®^ Reduced Serum Medium (Gibco, Thermo Fisher Scientific, Waltham, MA, USA) to a final volume of 125 μL. Lipofectamine 3000 (Invitrogen, Thermo Fisher Scientific, USA) was mixed with Opti-MEM^®^ in a 1:1 (*v*/*v*) and incubated at room temperature for 5 min. The plasmid and Lipofectamine 3000 solutions were combined, incubated for 15 min to allow complex formation, and then added dropwise to the cells. After 24 h, the cells were incubated with Mito Tracker Deep Red FM (Beyotime, Shanghai, China) at a concentration of 100 nM in serum-free DMEM for 30 min at 37 °C while protected from light. Cells were washed twice with 1 × PBS, stained with Hoechst 33342 (Lablead, Beijing, China) at 5 μg/mL for 10 min, washed again with 1 × PBS, and immediately imaged using a Leica TCS SP8 STED confocal microscope (Leica Microsystems, Wetzlar, Germany). Fluorescence signals were captured with the following settings: GFP (EGFP-MfpA): Ex/Em = 488–540 nm; Mito Tracker Deep Red FM: Ex/Em = 633–720 nm; Hoechst 33342: Ex/Em = 405–480 nm. Images were processed using Leica LAS X software (v3.7.4).

### 2.8. Detection of Apoptosis

The apoptosis rate of 48 h-infected Raw264.7 macrophages was assessed using the Annexin V/PI method (Annexin V-FITC Apoptosis Detection Kit, Beyotime, Shanghai, China). The control groups included a blank control group (uninfected cells in complete medium), an Annexin V single-staining group (cells stained with Annexin V-FITC only), and a propidium iodide (PI) single-staining group (cells stained with PI only). The cells were washed with PBS and digested with trypsin (Biosharp, Anhui, China) for 2 min, and the digestion was stopped with complete medium. After centrifuging at 1000× *g* for 5 min, we discarded the supernatant; then, we collected the cells, gently resuspended them in PBS, and counted them. We took ~1 × 10^4^ resuspended cells, centrifuged them at 1000× *g* for 5 min, discarded the supernatant, and added 195 μL Annexin V-FITC conjugate to gently resuspend the cells. We then added 5 μL Annexin V-FITC and/or 10 μL PI staining solution according to the set grouping and mixed it gently. The solution was incubated at room temperature (25 °C) for 20 min away from light, then placed on ice and analyzed by CytoFLEX (Beckman Coulter, CA, USA) using CytExpert software (v2.6, Beckman Coulter, USA).

### 2.9. Transmission Electron Microscopy

Monolayers of cells were grown on Aclar Embedding Film (Electron Microscopy Sciences, Hatfield, PA, USA). The cells were fixed with 2.5% (*v*/*v*) glutaraldehyde (Sigma-Aldrich, 111-30-8, St. Louis, MO, USA) in phosphate buffer (PB) (0.1 M, pH 7.4) at 4 °C, followed by two washes in PB and two washes in deionized water. Subsequently, the cells were immersed in a 1% (*w*/*v*) OsO_4_ and 1.5% (*w*/*v*) potassium ferricyanide aqueous solution (Sigma-Aldrich, St. Louis, MO, USA) at 4 °C for 1 h. After washing, the cells were dehydrated through a graded alcohol series (30%, 50%, 70%, 80%, 90%, 100%, and 100%, 5 min each at 4 °C), and then washed twice in pure acetone. The samples were infiltrated with a graded mixture of acetone and Embed 812 resin—containing 20 mL of 812 (Electron Microscopy Science, USA), 9 mL of DDSA-Dodecenyl Succinic Anhydride Specially Distilled (Electron Microscopy Science, USA), and 10 mL of NMA-Methy-5-Norbornene-2,3-Dicarboxylic Anhydride (Electron Microscopy Science, USA)—in 3:1, 1:1, and 1:3 ratios, followed by pure resin. The samples were embedded in pure resin containing 1.5% BDMA (N, N-Dimethylbenzylamine, SPI-CHEM, West Chester, PA, USA) and polymerized at 45 °C for 12 h and at 60 °C for 48 h. Ultrathin sections (70 nm) were cut using a Leica EM UC6 microtome (Leica Microsystems, Wetzlar, Germany) and double-stained with uranyl acetate and lead citrate. Lastly, the morphological features were observed using a Hitachi HT7800 TEM/Regulus8100 electron microscope (Hitachi LTD; Tokyo, Japan).

### 2.10. Determination of Mitochondrial Membrane Potential (ΔΨm) in Host Cells

The mitochondrial membrane potential (ΔΨm) of Raw264.7 macrophages infected for 48 h was assayed using a JC-1 fluorescent probe (Enhanced mitochondrial membrane potential assay kit with JC-1, Beyotime, Shanghai, China). Approximately 1 × 10^4^ cells were collected and resuspended in 0.5 mL of cell culture medium, followed by incubation with 0.5 mL of JC-1 staining working solution for 20 min in a cell culture incubator at 37 °C. After incubation, the cells were centrifuged at 600× *g* for 4 min at 4 °C and then precipitated, and the supernatant was discarded. The cells were washed twice with JC-1 staining buffer at 600× *g* for 4 min at 4 °C, and the supernatant was discarded. After resuspension in an appropriate amount of JC-1 staining buffer, the cells were analyzed using a CytoFLEX flow cytometer (Beckman Coulter, USA) and CytExpert software (Beckman Coulter, USA).

### 2.11. Detection of Reactive Oxygen Species (ROS) and Mitochondrial ROS in Host Cells

The ROS levels in Raw264.7 macrophages infected for 24 and 48 h were measured using a DCFH-DA fluorescent probe (Reactive Oxygen Detection Kit, Biotronik, Hangzhou, China). Approximately 1 × 10^4^ cells were collected, resuspended in 0.5 mL of ice-cold 1 × PBS, and incubated with 10 μM DCFH-DA for 40 min in the dark at 37 °C.

The mitochondrial ROS levels in Raw264.7 macrophages infected for 24 and 48 h were detected using a MitoSOX Red probe (Thermo Fisher Scientific, M36007, Waltham, MA, USA). Approximately 1 × 10^4^ cells were gently resuspended in a pre-warmed (37 °C) staining solution containing 250 nM MitoSOX Red probe and incubated in a cell culture incubator, protected from light, for 40 min. The ROS and mitochondrial ROS levels were analyzed using a CytoFLEX flow cytometer (Beckman Coulter, Brea, CA, USA) and FlowJo^®^ (version 10.8.1; FlowJo LLC., Ashland, OR, USA).

### 2.12. RNA Sequencing (RNA-Seq) Analysis and Validation of Recombinant M. bovis BCG Strains Following Infection of Host Cells and RT-qPCR

After infecting Raw264.7 macrophages for 48 h, the total RNA was extracted using TRIzol reagent (Invitrogen, CN, Carlsbad, CA, USA) according to the manufacturer’s instructions. RNA quality was assessed by agarose gel electrophoresis (OD_260_/_280_ ≥ 1.9), and RNA concentration and purity were measured using a Nanodrop Spectrophotometer (TIENGEN, Beijing, China). mRNA was then purified using oligo dT microbeads, fragmented, and reverse-transcribed to synthesize double-stranded cDNA. The cDNA was subjected to end repair, 3’ end ‘A’ addition, and adapter ligation for the construction of a sequencing library. High-fidelity polymerase amplification was performed to generate the sequencing library. The library was denatured with sodium hydroxide to form single-stranded DNA fragments, which were fixed on a microarray for bridge PCR amplification. DNA clusters were generated, linearized into single strands, and sequenced using the Illumina platform (San Diego, CA, USA). Three biological replicates were used for each sample, resulting in the sequencing of 12 transcriptome samples, generating 83.55 G of clean data. The effective data volume for each sample ranged from 6.88 to 7.06 G, with the Q30 base percentages ranging from 95.75 to 96.27%. The average GC content was 47.66% and the genome alignment rates ranged from 92.40% to 93.39%.

cDNA synthesis was performed using 5 × All-In One RT MasterMix (Applied Biological Materials, G592, Richmond, BC, Canada). Quantitative real-time PCR was performed using BlasTaqTM 2 × qPCR MasterMix (Applied Biological Materials, G891, Richmond, BC, Canada) on a Bio-Rad CFX Connect Real-Time SystemTM (Bio-Rad Laboratories, Hercules, CA, USA). *β*-*actin* was used as the housekeeping gene, and the relative gene expression was calculated using the 2^−∆∆CT^ method [20]. The RT-qPCR primers used are listed in Appendix A.

### 2.13. Cytokine Production Induced by Recombinant M. bovis BCG Strains in Raw264.7 Cells

The concentrations of TNF-α and IL-6 in the supernatants of Raw264.7 macrophages infected for 48 h were measured using sandwich enzyme-linked immunosorbent assay (ELISA) kits (UpingBio, Shanghai, China; TNF-α: SYP-M0036; IL-6: SYP-M0031). After 48 h of infection, cell culture supernatants were harvested and centrifuged at 2000× *g* for 10 min at 4 °C to remove cell debris. The supernatants were aliquoted and stored at −80 °C until analysis. Lyophilized TNF-α and IL-6 standards were reconstituted in assay diluent (provided in the kit) to generate a 7-point standard curve. Serial dilutions were prepared in sterile PBS, and pre-coated 96-well plates were equilibrated to room temperature (25 °C) for 30 min.

For the ELISA procedure, the following wells were set up: calibrator, sample dilution, blank, and sample wells. Fifty microliters of calibrator, at various concentrations, was added to the calibrator wells, while the blank well was left empty. Fifty microliters of sample dilution and 50 μL of the supernatant were added to the sample dilution and sample wells, respectively. In all wells except the blank, 100 μL of biotin-labeled antibody was added, and the plates were incubated in the dark at 37 °C for 60 min. The plates were then washed 5 times with wash buffer. HRP-labeled avidin (100 μL) was added to each well (except the blank), and the plates were incubated for 20 min at 37 °C in the dark. After washing the plates 5 times, 100 μL of TMB substrate solution was added to each well, and the optical density (OD) was measured at 450 nm using a SpectraMax^®^ Paradigm Multi-Mode Microplate Reader (Molecular Devices, San Jose, CA, USA). Standard curves were generated using 4-parameter logistic (4PL) regression in ELISA Calc v0.2 (Molecular Devices).

### 2.14. Western-Blotting

After 48 h of infection, Raw264.7 macrophages were washed with 1 x PBS, then centrifuged at 12,000× *g* for 5 min at 4 °C, and the supernatant was discarded. Cells were lysed with cell lysis buffer (Lablead, Beijing, China) containing protease inhibitor (1 mM PMSF (Lablead, Beijing, China) and a 1% phosphatase inhibitor cocktail (Lablead, Beijing, China), followed by 30 min of incubation on ice. The lysates were centrifugated at 12,000× *g* for 5 min at 4 °C, and the supernatant was collected. The protein concentration was measured, and the samples were mixed with protein loading buffer and boiled at 95 °C for 5 min. SDS-PAGE was performed to separate the proteins, which were then transferred to a polyvinylidene difluoride (PVDF) membrane (Millipore immobilon-P, Billerica, MA, USA).

The membrane was blocked with 5% skim milk (Becton Dickinson, Sparks, MD, USA) for 2 h at room temperature, and was then incubated with primary antibodies against Bax (1:1000, Abmart, Shanghai, China), Bcl2 (1:1000, Abmart, T40056 China), β-actin (1:8000, Abmart, P30002, China), β-tublin (1:8000, Abmart, M20005, China), Gadd45B (1:1000, Bioss, bs-5904R, China), p65 (1:1000, Abmart, T55034, China), phospho-p65 (1:1000, Abmart, TP56372, China), IκBα (1:1000, Abmart, T55026, China), and phospho-IκBα (1:1000, Abmart, TA2002, China). After washing with PBST (1 × PBS containing 0.05% (*v*/*v*) Tween 80 (Sinopharm Chemical Reagent, Shanghai, China), the membranes were incubated with horseradish peroxidase (HRP)-conjugated secondary antibody goat anti-rabbit IgG (1:10,000) or goat anti-mouse IgG (1:10,000) (Cell Signaling Technology, Danvers, MA, USA). The protein bands were visualized using enhanced chemiluminescence (ECL) substrate (Thermo Fisher Scientific), and the images were analyzed using ImageJ (v1.54p, National Institutes of Health, Bethesda, MD, USA).

### 2.15. Statistical Analysis

All experiments were independently repeated at least three times. The data are expressed as the mean ± standard error of the mean (SEM). For comparisons between two groups, the normality of data distribution was first assessed using the Shapiro–Wilk test (for small sample sizes, *n* ≤ 50) or the D’Agostino–Pearson test (for large sample sizes, n > 50). If data passed the normality test (*p* > 0.05), parametric tests (unpaired Student’s *t*-test for equal variances or Welch’s *t*-test for unequal variances) were applied. If data did not pass the normality test, the non-parametric Mann–Whitney U test was used. For comparisons involving three or more groups, normality was verified via the Shapiro–Wilk test (*p* > 0.05), and homogeneity of variances was confirmed by the Brown–Forsythe test (*p* > 0.05). One-way ANOVA followed by Tukey’s post hoc test was performed for parametric data. Statistical analyses were performed using GraphPad Prism v9.0 (GraphPad Software, San Diego, CA, USA). The significance of the *p* values was determined as follows: **** *p* < 0.0001; *** *p* < 0.001; ** *p* < 0.01; * *p* < 0.05; and ns, not significant.

## 3. Results

### 3.1. Knockout of the Gene Encoding the Antibiotic-Resistant Protein (ARP) MfpA Significantly Reduced Mycobacterial Survival Within Host Cells

Drug-resistance proteins in Mtb play roles in both virulence and antibiotic susceptibility, contributing to the bacteria’s ability to cause disease while influencing its resistance to treatment. However, the underlying mechanisms of these roles remain poorly understood. To explore whether MfpA contributes to mycobacterial virulence beyond its established role in fluoroquinolone resistance, we first examined the effect of MfpA on Mtb virulence. In this study, we used *M. bovis* BCG, a safer, attenuated strain of *M. bovis* that shares genetic similarities with Mtb, instead of the virulent Mtb strain, to investigate MfpA’s role in virulence. Using a specialized transduction approach [21,22], we generated an *mfpA*-knockout strain named BCG::Δ*mfpA* (Figure 1A,B). We then assessed whether *mfpA* knockout affected mycobacterial fitness by measuring the growth curve in vitro using a 7H9 medium supplement of ADS. The growth kinetics experiment revealed no significant differences in the growth rates between the wild-type strain and the *mfpA*-knockout, complemented, and overexpression strains, suggesting that MfpA does not influence the in vitro growth of *M. bovis* BCG (Figure 1B).

To investigate the role of MfpA during *M. bovis* BCG invasion in host cells, we performed the infection experiments using Raw264.7 cells with WT BCG, BCG::Δ*mfpA*, pMV361-*mfpA*/BCG::Δ*mfpA*, and pMV261-*mfpA*::BCG at a multiplicity of infection (MOI) of 5. Intracellular colony counting showed that the colony-forming units (CFUs/mL) of BCG::Δ*mfpA* were significantly lower than those of WT BCG at 48 h post-infection (hpi). However, the CFUs of pMV361-*mfpA*/BCG::Δ*mfpA* were restored to levels comparable to those of WT BCG. Additionally, overexpression of *mfpA* in the pMV261-*mfpA*::BCG strain resulted in a significant increase in CFUs (Figure 2A). These results suggest that MfpA plays a role in the survival of Mtb within host cells.

### 3.2. Knockout of the Gene Encoding the ARP MfpA duced Apoptosis and Pro-Inflammatory Cytokine Production

In addition, to assess the expression of inflammatory factors during the infection of Ras264.7 macrophages with WT BCG, BCG::Δ*mfpA*, pMV361-*mfpA*/BCG::Δ*mfpA*, and pMV261-*mfpA*::BCG, we measured the levels of TNF-α and IL-6 in the cell culture supernatants. The results revealed that MfpA deletion significantly increased the expression of TNF-α and IL-6 (Figure 2B). This suggests that MfpA plays a role in inhibiting the production of these pro-inflammatory cytokines. These findings indicate that MfpA significantly enhances the intracellular survival of BCG in macrophages, promoting the survival of mycobacterial strains within these cells.

Macrophage-mediated natural immunity is the first line of host defense against Mtb, and apoptosis is a critical defense mechanism that helps control pathogen invasion and bacterial growth [23]. To further investigate the mechanism by which MfpA promotes intracellular mycobacterial survival, we infected Raw264.7 with WT BCG, BCG::Δ*mfpA*, and pMV361-*mfpA*/BCG::Δ*mfpA* for 48 h, followed by flow cytometry analysis. Compared to the control group (cell culture medium as a blank), infection with WT BCG led to a slight increase in apoptotic macrophages. As anticipated, macrophages infected with the *mfpA*-knockout strain BCG::Δ*mfpA* showed a significant increase in the proportion of apoptotic cells compared to the WT BCG-infected group. However, the proportion of apoptotic macrophages was restored to a level similar to that of the WT BCG-infected group when the corresponding complemented strain (pMV361-*mfpA*/BCG::Δ*mfpA*) was used (Figure 3A). In parallel, Western-blotting analysis of Bax and Bcl2 expression revealed a significantly higher Bax/Bcl2 ratio in macrophages infected with BCG::Δ*mfpA* compared to both the control and WT BCG-infected groups (Figure 3B). This elevated Bax/Bcl2 ratio indicated enhanced pro-apoptotic signaling in the absence of MfpA, suggesting that MfpA normally suppresses macrophage apoptosis. Furthermore, ultrastructural analysis showed that Raw264.7 cells infected with the *mfpA*-knockout strain BCG::Δ*mfpA* exhibited a higher incidence of nuclear pyknosis and fragmentation, cytoplasmic condensation, organelle swelling, and vacuolization (Figure 3C). The quantitative analysis of apoptotic cell ratios from six fields of view per group showed the uninfected control (Ctrl) with ~15% ± SEM apoptosis. The BCG::Δ*mfpA*-infected group (ΔMfpA) had a significantly higher ratio of ~70% ± SEM (**** *p* < 0.0001 *vs.* Ctrl), indicating increased apoptosis without MfpA. The pMV361-*mfpA*/BCG::Δ*mfpA*-infected group (MfpA + ΔMfpA) exhibited a reduced ratio of ~40% ± SEM (**** *p* < 0.0001 *vs.* ΔMfpA; **** *p* < 0.0001 *vs.* Ctrl). These findings suggest that MfpA plays a crucial role in inhibiting macrophage apoptosis, thereby promoting the intracellular survival of *Mycobacterium*.

### 3.3. The ARP MfpA Targeted Mitochondria and Dysregulated Mitochondrial Function

Knocking out MfpA did not affect bacterial growth in vitro (Figure 1C). The above study demonstrated that MfpA regulates the expression of TNF-α and IL-6 and affects apoptosis, suggesting that MfpA is a virulence factor. Previous studies showed that MfpA was identified in membrane fractions, which suggests that it may be secreted through a membrane-associated transport system [24,25]. Additionally, within the MfpA (Rv3361c) operon, Rv3365c has been identified as a secretion-related protein [26], further suggesting that MfpA (Rv3361c) could be a secreted protein influencing host cells. To test this hypothesis, MfpA was fused with a FLAG tag at the N-terminus and expressed under the control of the heat shock protein (*hsp*) promoter using the pMV261 multicopy vector [27]. Positive clones were selected and cultured under shaking conditions to expand the bacterial culture into the logarithmic phase. Both the supernatant and bacterial proteins were then collected separately. The FLAG-tagged MfpA protein was detected in both the supernatant and the bacterial cells (Figure 4A), suggesting that MfpA is secreted by the bacteria.

Although previous studies on MfpA, including our own, primarily focused on its role in conferring resistance to FQs in *Mycobacterium* [28,29], the observation of its exocytosis prompted us to investigate the underlying mechanisms of MfpA secretion and its effect on host cells. To explore the cellular localization of MfpA further, we mimicked the process of bacterial infection and protein secretion into the host environment. We constructed a eukaryotic expression vector, pcDNA3.1(+), to fuse MfpA with green fluorescent protein (GFP), which was then transfected into HEK-293T cells. Confocal microscopy was used to examine the intracellular distribution of MfpA, with nuclear (Hoechst 33343) and mitochondrial (Mito Tracker Deep Red FM) staining aiding in its localization within the cell. The results showed that MfpA was distributed throughout the cytoplasm, with partial overlap in areas associated with the mitochondria, suggesting its potential involvement in host cell interactions during infection (Figure 4B).

Mitochondria are recognized as critical regulators of both antimicrobial defense and cell death. Given that MfpA localized to the mitochondria of host cells, we hypothesized that MfpA may affect mitochondrial function. Mitochondria not only are the powerhouses of cells, responsible for energy production, but also serve as their metabolic center. The mitochondrial respiratory chain establishes a membrane potential (ΔΨm) across the inner mitochondrial membrane via electron transfer. ΔΨm is essential for various cellular processes, including ATP synthesis and the transport of mitochondrial precursor proteins and metabolites across the inner membrane [30]. To test our hypothesis, we assessed the ΔΨm in host cells infected with WT BCG, BCG::Δ*mfpA*, and pMV361-*mfpA*/BCG::Δ*mfpA* for 48 h. The results revealed that infection with BCG::Δ*mfpA* resulted in a higher proportion of JC-1 monomeric forms in host cells, reflected by a stronger green fluorescent signal, which suggested a decrease in mitochondrial membrane potential and less healthy cells. This finding corresponded to a significant reduction in ΔΨm in the BCG::Δ*mfpA*-infected group compared to both the WT BCG-infected group and the complemented strain, pMV361-*mfpA*/BCG::Δ*mfpA* (Figure 5).

Reactive oxygen species (ROS) are crucial in the development and progression of tuberculosis. During infection, Mtb encounters ROS, such as superoxide anions (O_2_^−^), hydrogen peroxide (H_2_O_2_), and hydroxyl radicals (-OH), produced by the host’s immune system. In order to survive and replicate within host macrophages, Mtb must effectively counteract these ROS. To explore the role of MfpA in ROS modulation, we measured ROS levels in cells infected with WT BCG, BCG::Δ*mfpA*, and pMV361-*mfpA*/BCG::Δ*mfpA* for 24 h and 48 h. The results showed a significant increase in ROS levels in the BCG::Δ*mfpA-*infected group, while the complemented strains exhibited ROS levels similar to those of the WT BCG-infected group (Figure 6A). To further investigate the effect of MfpA on mitochondrial function, we employed the fluorescent probe MitoSOX Red, which specifically detects superoxide within mitochondria. As expected, the results revealed a marked increase in mitochondrial ROS levels in the BCG::Δ*mfpA*-infected group, supporting our hypothesis (Figure 6B).

### 3.4. The ARP MfpA Modulates the NF-κB Signaling Pathway and GADD45B Expression to Affect Host Apoptosis

To further investigate the processes and mechanisms by which MfpA regulates host cells during mycobacterial infections, we performed RNA sequencing (RNA-seq) on cells infected with WT BCG, BCG::Δ*mfpA*, and pMV361-*mfpA*/BCG::Δ*mfpA* for 48 h. Compared to the WT BCG-infected group, we identified a series of differentially expressed genes, most of which were involved in host infection and defense responses. To validate the reliability of the RNA-seq results, we performed RT-qPCR, and the results confirmed consistency between the RNA-seq and RT-qPCR data (Figure 7). Notably, we observed a significant increase in the transcript levels of the *gadd45β* gene in BCG::Δ*mfpA*-infected macrophages compared to the WT BCG-infected group. Gadd45β, a member of the Gadd45 family, is known to regulate various biological processes through protein–protein interactions in both the nucleus and cytoplasm [31]. However, despite the elevated transcript levels, GADD45B protein expression was significantly reduced in BCG::Δ*mfpA*-infected macrophages, as confirmed by the specific antibody assay (Figure 8).

Moreover, pathway enrichment analysis revealed significant enrichment in the Kyoto Encyclopedia of Genes and Genomes (KEGG) and Gene Ontology (GO) pathways in relation to apoptosis in the BCG::Δ*mfpA*-infected group, suggesting that MfpA may influence the modulation of the NF-κB signaling pathway in host cells (Figure 9). To further investigate this, we examined the expression levels of NF-κB p65, IkBα, and their respective phosphorylated forms in WT BCG-, BCG::Δ*mfpA-*, and pMV361-*mfpA*/BCG::Δ*mfpA*-infected Raw264.7 cells. Infection with BCG::Δ*mfpA* resulted in altered phosphorylation of both NF-κB p65 and IkBα compared to the WT BCG-infected group (Figure 10).

In summary, our study highlights the complex role of MfpA in regulating host cell responses during *Mycobacterium* infection, particularly through alterations in key signaling pathways such as apoptosis and the activation of the NF-κB signaling pathway.

## 4. Discussion

In this study, we revealed a previously unrecognized dual role of the mycobacterial antibiotic resistance protein (ARP) MfpA; we achieved this using *Mycobacterium bovis Bacillus Calmette-Guérin* (*M. bovis* BCG), a widely used vaccine strain related to *Mycobacterium tuberculosis* (Mtb), which is a model organism for studying mycobacterial survival mechanisms. While MfpA is well known for its role as a resistance factor, we show for the first time that it also functions as a virulence factor that enhances mycobacterial survival. It achieves this by modulating host mitochondrial function and suppressing apoptosis via the NF-κB signaling pathway. This novel mechanism contributes to *M. bovis* BCG’s ability to evade host immune responses and persist within host cells. Our finding provides the first direct evidence of MfpA’s interaction with host mitochondria, affecting key cellular processes such as reactive oxygen species (ROS) generation and mitochondrial membrane potential (ΔΨm), both crucial for apoptosis induction.

The link between antibiotic resistance and virulence in mycobacteria has been sporadically reported [11,32,33]; however, our work is the first to show that MfpA—an ARP associated with fluoroquinolone resistance through DNA gyrase protection—directly manipulates host mitochondrial functions to suppress apoptosis (Figure 11). This shifts the perspective on antibiotic resistance genes, positioning them as active participants in immune evasion rather than survival tools against antibiotics. The reduced survival of the Δ*mfpA*::BCG strain in host cells (Figure 2A), coupled with elevated TNF-α/IL-6 levels and increased apoptosis (Figure 2B and Figure 3), emphasizes the crucial role of MfpA in both virulence and resistance.

Mitochondria play a crucial role in regulating apoptosis, a process that eliminates infected or damaged cells for immune defense. However, pathogens have evolved mechanisms to evade apoptosis and prolong their survival within host cells. In this study, we showed that MfpA co-localized with host mitochondria (Figure 4B), affecting ROS production and mitochondrial membrane potential (ΔΨm)—key triggers for apoptosis [34]—indicating that MfpA targets mitochondria to block apoptosis (Figure 3C).

The manipulation of mitochondrial ROS is closely linked to pro-inflammatory signaling through the activation of the NF-κB signaling pathway, which drives cytokine production [35,36]. This immune evasion strategy is similar to mechanisms employed by other pathogens. For example, *Legionella pneumophila* stabilized mitochondrial membranes through its MitF protein to inhibit apoptosis, and *Salmonella* used its SseL protein to degrade mitochondrial ubiquitin chains, suppressing ROS-driven NF-κB signaling pathway activation [35,37]. Similarly, Mtb proteins like Rv1813c [38] and Rv0674 [8] also target host mitochondria to induce apoptosis. These studies highlight mitochondria as a crucial site for both pathogens and hosts, and suggest that host-directed therapy holds promise for TB management. For example, metformin, a hypoglycemic drug that inhibits mitochondrial complex I, has been shown to reduce the risk of TB [39]. In our study, the drug-resistance-associated gene *mfpA* was found to target host mitochondria in *M. bovis* BCG, suggesting that mitochondria-targeting drugs hold promise for addressing the emerging issue of drug resistance in infectious diseases. Additional research is necessary to fully understand these interactions and to identify new therapeutic targets for combating Mtb and similar pathogens.

The NF-κB signaling pathway plays a dual role in tuberculosis (TB) by promoting antibacterial defense while also potentially causing harm to the host through excessive inflammation [40,41]. MfpA was found to modulate NF-κB activity, paradoxically reducing apoptosis despite NF-κB’s canonical role in promoting anti-apoptotic responses [42,43]. This paradox may arise because, in the context of BCG infection, suppression of the NF-κB pathway mitigates excessive TNF-α-driven inflammation, thereby preventing inflammation-induced apoptosis. Additionally, by stabilizing IκB, MfpA may counteract JNK-mediated apoptosis, contributing to the survival of infected cells and promoting bacterial persistence. In addition, we also observed that inhibiting NF-κB signaling affects the expression of GADD45B, a key protein involved in DNA damage repair, cell cycle regulation, and apoptosis [44,45]. While *gadd45β* expression was upregulated at the transcriptional level (Figure 8A), its protein levels were significantly reduced in the BCG::Δ*mfpA*-infected group (Figure 8B). This suggest that MfpA may modulate GADD45B protein levels through the NF-κB suppression pathway, potentially affecting its translation or stability. These changes could affect the host’s immune response and alter the outcome of infection.

MfpA’s multifunctional role both in antibiotic resistance and as a virulence factor offers promising opportunities for combination therapies targeting both pathogen resistance and virulence mechanisms. Such a dual-target approach could reduce “single-pressure selection”, delay drug resistance, and improve treatment effectiveness. Inhibiting MfpA’s virulence factors may restore immune functions suppressed by MfpA, such as ROS production and apoptosis inhibition, thereby strengthening the immune response and improving the efficacy of existing antibiotics. This could ultimately lead to faster bacterial clearance, reduced relapse, and delayed resistance development, offering a more sustainable treatment for tuberculosis.

Our study also unlocks exciting possibilities for therapeutic interventions aimed at reducing pathogen virulence while simultaneously restoring normal immune function. For example, compounds designed to reverse the effects of MfpA on mitochondrial ROS production or apoptotic inhibition could enhance host immune responses and improve clinical outcomes in Mtb infections. Combination therapies targeting both mitochondrial dynamics and immune modulation could harness the full potential of the host immune system, offering a more holistic and effective approach to treatment.

Furthermore, the involvement of GADD45B in modulating pro-inflammatory cytokines and immune cell activation suggests that it could serve as a valuable therapeutic target. Understanding the precise mechanisms underlying GADD45B’s role and its interactions with other immune pathways will be crucial for optimizing drug development.

In conclusion, this work highlights the importance of considering antibiotic resistance proteins as multifunctional virulence factors. By revealing how MfpA actively modulates host cellular mechanisms, we unlock new avenues for therapeutic intervention. Targeting MfpA’s interactions with host mitochondria or its effects on NF-kB signaling could provide strategies to enhance immune responses while combating antibiotic resistance in Mtb. Further research should aim to explore additional ARPs that share similar functions and refine therapeutic strategies to disrupt these pathways, ultimately improving treatment outcomes for tuberculosis and similar infections.

## Figures and Tables

**Figure 1 cells-14-00867-f001:**
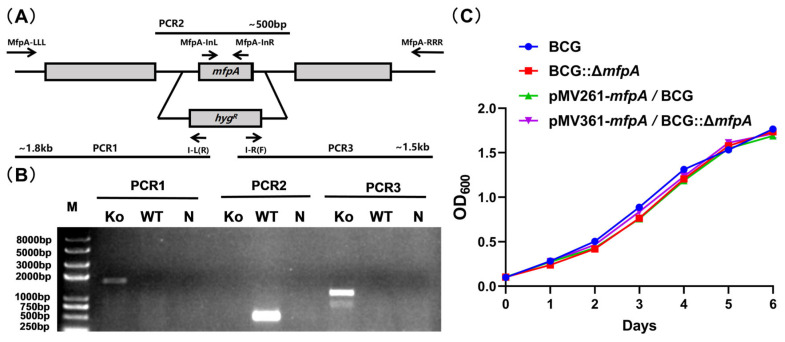
The effect of MfpA on mycobacterial survival in vitro. (**A**) *mfpA*-knockout strategy. A genomic region approximately 1 kb upstream and downstream of the *mfpA* gene was amplified by PCR, and an allelic exchange vector was constructed using phAE159. Positive clones were selected on Hygromycin-resistant plates, followed by PCR verification to confirm the substitution of *mfpA* with the hygromycin resistance gene (*hyg^R^*). (**B**) Screening and identification of knockout strains. The knockout strains were screened and identified by PCR. PCR1, PCR2, and PCR3 were performed using the primer pairs: MfpA-LLL and I-L(R), MfpA-InL and MfpA-InR, I-R(F) and MfpA-RRR, respectively. KO: *mfpA*-knockout strain; WT: wild-type strain; N: negative control. (**C**) In vitro growth curve. Complemented (pMV361-*mfpA*/BCG::Δ*mfpA*) and overexpression (pMV261-*mfpA*::BCG) strains were constructed. Bacterial growth was monitored under standard culture conditions by measuring the optical density at 600 nm (OD_600_) every 24 h over a 7-day period. The strains are represented as follows: BCG (blue filled oval); BCG::Δ*mfpA* (red square); pMV261-*mfpA*::BCG (green triangle); pMV361-*mfpA*/BCG::Δ*mfpA* (purple inverted triangle).

**Figure 2 cells-14-00867-f002:**
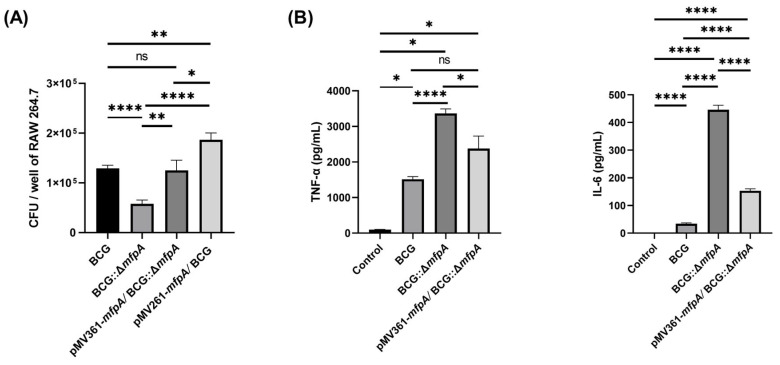
The effect of MfpA on the intracellular survival of *M. bovis* BCG in Raw264.7 macrophages. (**A**) The intracellular survival of different *M. bovis* BCG strains in Raw264.7 macrophages at 48 h post-infection (hpi). Macrophages were infected with WT BCG, BCG::Δ*mfpA*, pMV361-*mfpA*/BCG::Δ*mfpA*, and pMV261-*mfpA*::BCG at an MOI of 5. The plate counting of intracellular colony-forming units (CFUs/mL) showed a significant reduction in CFUs for BCG::Δ*mfpA* compared to WT BCG, indicating that MfpA deletion impaired bacterial survival. Complementation with pMV361-*mfpA* restored CFU levels to those of WT BCG, while the overexpression of MfpA in pMV261-*mfpA*::BCG further enhanced intracellular bacterial survival. These findings suggest that MfpA promotes the survival of *M. bovis* BCG within host macrophages. (**B**) The secretion of inflammatory cytokines TNF-α and IL-6 in the supernatants of Raw264.7 macrophages infected with different strains of *M. bovis* BCG. MfpA deletion in BCG::Δ*mfpA* resulted in significantly higher levels of TNF-α and IL-6 compared to WT BCG, suggesting that MfpA inhibits the production of these pro-inflammatory cytokines during infection. The data are presented as the mean ± SEM ad were analyzed using Student’s *t*-test. Significance was determined based on the *p* values (**** *p* < 0.0001; ** *p* < 0.01; * *p* < 0.05; ns, not significant).

**Figure 3 cells-14-00867-f003:**
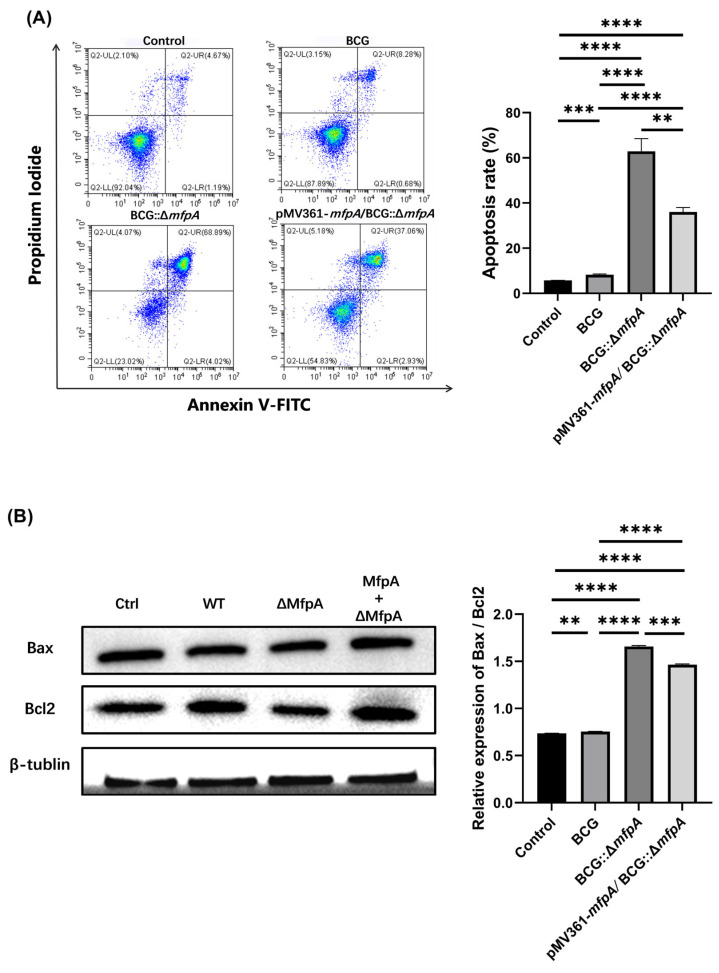
The effect of MfpA on macrophage apoptosis and its impact on mycobacterial survival. (**A**) Annexin V/PI staining was conducted to determine the apoptosis rate. Flow cytometry analysis of apoptotic macrophages 48 h post-infection with WT BCG, BCG::Δ*mfpA*, and pMV361-*mfpA*/BCG::Δ*mfpA*. The knockout of *mfpA* (BCG::Δ*mfpA*) significantly increased the proportion of apoptotic macrophages compared to WT BCG, while the complemented strain pMV361-*mfpA*/BCG::Δ*mfpA* restored apoptosis levels to those observed in the WT BCG-infected group. (**B**) Western-blotting analysis of Bax and Bcl2 expression in macrophages at 48 h post-infection. The Bax/Bcl2 ratio was significantly higher in macrophages infected with BCG::Δ*mfpA* compared to both the control and WT groups, suggesting enhanced apoptosis in the absence of MfpA. These results indicate that MfpA plays a crucial role in modulating macrophage apoptosis, thereby favoring the intracellular survival of *Mycobacterium*. A quantitative analysis of the results was performed by analyzing the grayscale intensity of the corresponding bands, followed by statistical analysis. β-tublin was used as a loading control. (**C**) A morphological assessment of MfpA’s effects on apoptosis in Raw 264.7 cells at 48 h post-infection. The blue arrow indicates cytoplasmic vacuolization in an apoptotic cell; the red arrow indicates the cell nucleus. Scale bar: 5 μm. The ratio of apoptotic cells was determined based on morphological characteristics in six fields of view per group. The data are presented as the mean ± SEM and were analyzed using Student’s *t-*test. Significance levels: **** *p* < 0.0001; *** *p* < 0.001; ** *p* < 0.01. Groups: Ctrl (uninfected control); WT (wild-type BCG-infected); ΔMfpA (BCG::Δ*mfpA*-infected); MfpA + ΔMfpA (pMV361-*mfpA*/BCG::Δ*mfpA*-infected).

**Figure 4 cells-14-00867-f004:**
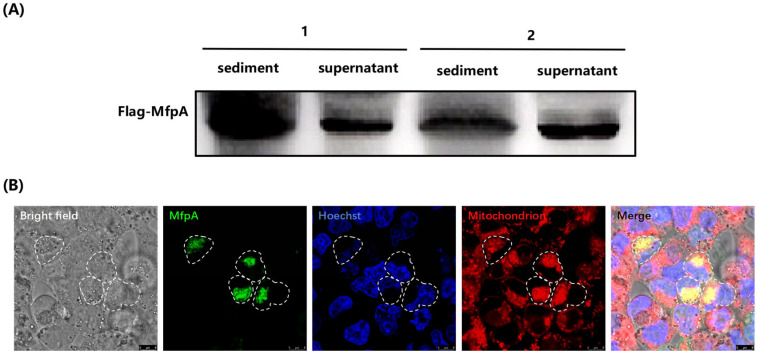
The secretion and mitochondrial localization of MfpA during bacterial growth. (**A**) The cell fractionation of FLAG-tagged MfpA in wild-type BCG. The *mfpA* gene was fused with a FLAG tag at the N-terminus and expressed under the *hsp* promoter using the pMV261 vector. Positive clones were cultured to the logarithmic phase. The culture supernatant and bacterial proteins were collected, precipitated, and separated on SDS-PAGE. FLAG-MfpA expression was determined in both the supernatant and bacterial fractions. Clones 1 and 2 represent independent isolates. (**B**) The localization of MfpA in HEK-293T cells. HEK-293T cells were transfected with a pcDNA3.1(+) vector containing MfpA fused to GFP. The cellular distribution of MfpA was visualized using Leica TCS SP8 STED 24 h post-transfection. Mitochondria and nuclei were stained with Mito Tracker Deep Red FM and Hoechst 33343, respectively. Green: MfpA; Blue: nuclei; Red: mitochondria; and the white dotted line represents the cell outline. Scale: 8 μm.

**Figure 5 cells-14-00867-f005:**
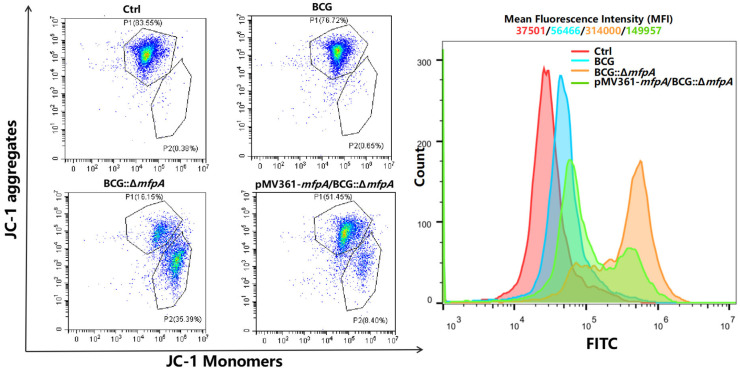
MfpA alters mitochondrial membrane potential (ΔΨm) in host cells. An assessment of ΔΨm was conducted 48 h post-infection with WT BCG, BCG::Δ*mfpA*, and pMV361-*mfpA*/BCG::Δ*mfpA*. The proportion of JC-1 monomeric forms was measured in the BCG::Δ*mfpA*-infected group and compared to that in the WT BCG-infected group. The left panel shows cellular compartmentalization, while the right panel visualizes the mean fluorescence intensity (MFI) in the FITC channel for each group. Red curve: Ctrl group (MFI: 37501); blue curve: BCG-infected group (MFI: 56466); orange curve: BCG::Δ*mfpA*-infected group (MFI: 314000); green curve: pMV361-*mfpA*/BCG::Δ*mfpA*-infected group (MFI: 149957).

**Figure 6 cells-14-00867-f006:**
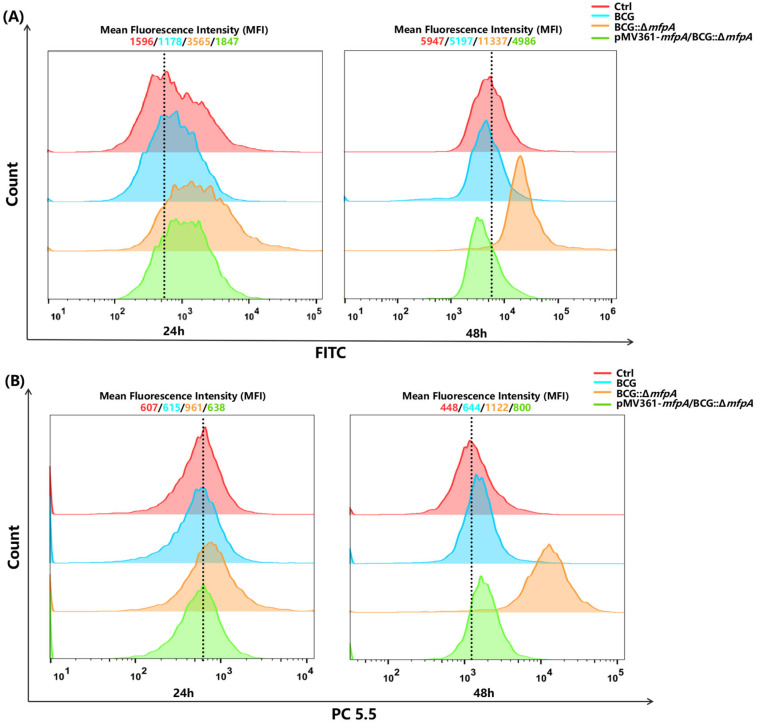
MfpA affects reactive oxygen species (ROS) production in host cells, particularly mitochondrial ROS. (**A**) ROS levels in host cells. ROS levels were measured 48 h post-infection with WT BCG, BCG::Δ*mfpA*, and pMV361-*mfpA*/BCG::Δ*mfpA.* (**B**) Mitochondrial ROS detection using MitoSOX Red. Mitochondrial ROS levels were measured 48 h post-infection with WT BCG, BCG::Δ*mfpA*, and pMV361-*mfpA*/BCG::Δ*mfpA*. The mean fluorescence intensities (MFIs) in the FITC or PC 5.5 channel for each group are represented as follows: red curve: Ctrl group; blue curve: BCG-infected group; orange curve: BCG::Δ*mfpA*-infected group; green curve: pMV361-*mfpA*/BCG::Δ*mfpA*-infected group.

**Figure 7 cells-14-00867-f007:**
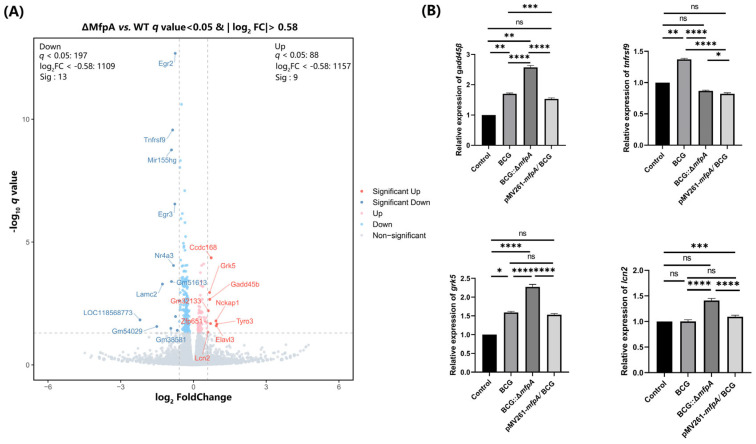
RNA-seq results and validation by RT-qPCR. (**A**) Differentially expressed genes from RNA-seq analysis. (**B**) RT-qPCR validation of gene expression in BCG::Δ*mfpA*-infected macrophages compared to BCG-infected macrophages. Data are presented as mean ± SEM, and were analyzed using Student’s *t*-test. Significance is indicated by *p* values (**** *p* < 0.0001; *** *p* < 0.001; ** *p* < 0.01; * *p* < 0.05; ns, not significant). WT, wild-type BCG-infected group; ΔMfpA, BCG::Δ*mfpA*-infected group.

**Figure 8 cells-14-00867-f008:**
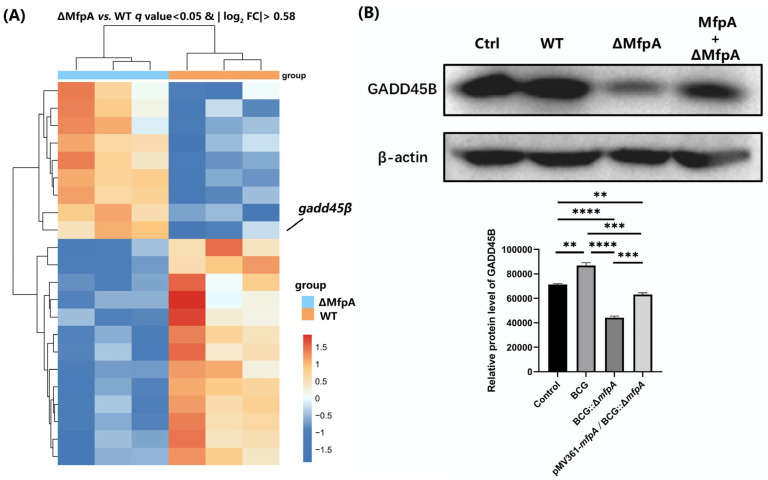
GADD45B protein expression in infected macrophages. (**A**) Transcript levels of *Gadd45β* were upregulated in BCG::Δ*mfpA*-infected cells. (**B**) Western-blotting analysis of GADD45B protein expression in macrophages infected with WT BCG, BCG::Δ*mfpA*, and pMV361-*mfpA*/BCG::Δ*mfpA*. Quantitative test was performed by analyzing grayscale intensity of the corresponding bands. β-actin was used as loading control. Data are presented as mean ± SEM, and were analyzed using Student’s *t-*test. Significance is indicated by *p* values (**** *p* < 0.0001; *** *p* < 0.001; ** *p* < 0.01;). Ctrl: uninfected control; WT, wild-type BCG-infected group; ΔMfpA, BCG::Δ*mfpA*-infected group; MfpA + ΔMfpA, pMV361-*mfpA*/BCG::Δ*mfpA*-infected group.

**Figure 9 cells-14-00867-f009:**
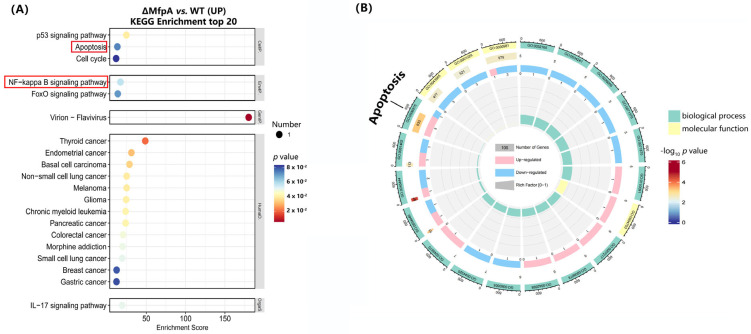
The enrichment analysis revealed potential effects of MfpA on the NF-κB signaling pathway and apoptosis in host cells. (**A**) The Kyoto Encyclopedia of Genes and Genomes (KEGG) pathway and (**B**) Gene Ontology (GO) enrichment analyses. KEGG pathway (**A**) and GO (**B**) enrichment analyses were performed on WT BCG and BCG::Δ*mfpA*-infected cells to identify significantly enriched pathways. Red frames highlight the pathways with the most notable enrichment in BCG::Δ*mfpA*-infected cells. WT, wild-type BCG-infected group; ΔMfpA, BCG::Δ*mfpA*-infected group.

**Figure 10 cells-14-00867-f010:**
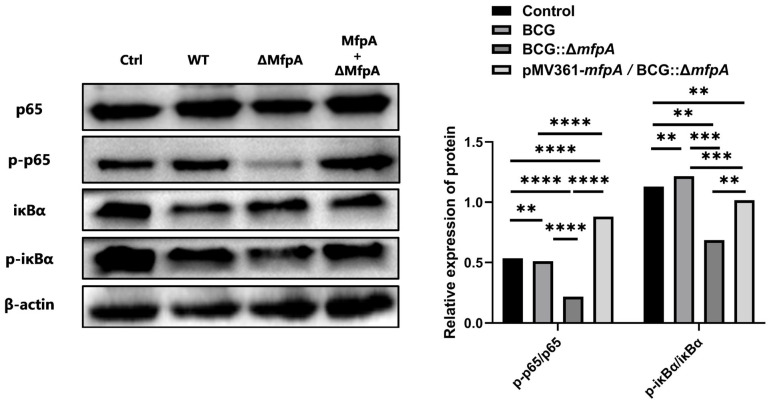
MfpA modulates the NF-κB signaling pathway in host cells. Western-blotting analysis of NF-κB p65, IκBα, and their phosphorylation forms in Raw264.7 cells 48 h post-infection with WT BCG, BCG::Δ*mfpA*, and pMV361-*mfpA*/BCG::Δ*mfpA*. Quantification was performed by analyzing the grayscale intensity of the corresponding bands, and β-actin was used as a loading control. The data are presented as the mean ± SEM, and were analyzed by one-way ANOVA. Significance is indicated by *p* values (**** *p* < 0.0001; *** *p* < 0.001; ** *p* < 0.01). Ctrl, uninfected control; WT, wild-type BCG-infected group; ΔMfpA, BCG::Δ*mfpA*-infected group; MfpA + ΔMfpA, pMV361-*mfpA*/BCG::Δ*mfpA*-infected group.

**Figure 11 cells-14-00867-f011:**
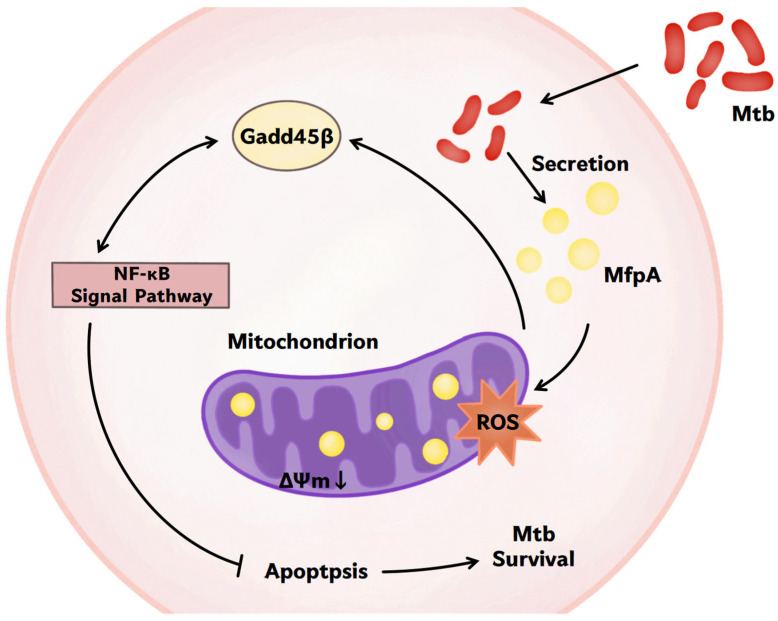
Proposed model of MfpA’s interaction with macrophages. MfpA, an antibiotic-resistant protein (ARP) associated with fluoroquinolone resistance, modulates host mitochondrial function to inhibit apoptosis and promote mycobacterial survival through the NF-κB signaling pathway.

## Data Availability

The authors declare that all of the data supporting the findings of this study are available within the article and its Appendix A or from the corresponding author upon reasonable request. The source data underlying Figure 1, Figure 2, Figure 3, Figure 7, Figure 8 and Figure 10 are provided in a Source Data file.

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
