# Peer review of "The Antibiotic-Resistant Protein MfpA Modulates Host Cell Apoptosis and Promotes Mycobacterial Survival by Targeting Mitochondria and Regulating the NF-κB Signaling Pathway"

_cells, 2025, doi:10.3390/cells14120867_

Round 1

Reviewer 1 Report

Comments and Suggestions for Authors

The manuscript presents a novel and well-structured investigation of the MfpA protein, linking its antibiotic resistance function to modulation of host mitochondrial dynamics and immune evasion in Mycobacterium bovis BCG infection. The study is relevant, particularly in the context of persistent tuberculosis (TB) infections and resistance development. I have two minor comments.

  1.  Line 84–86: The authors mentioned MfpA is secreted by the bacteria and localizes to host mitochondria. However, the mechanism of secretion is not described. Consider including known or hypothesized secretion pathways with appropriate citations.
  2. Line 91–92: "Promotes Mtb survival by modulating the NF-κB pathway." However, the data shows this conclusion is based on M. bovis BCG. Please avoid overgeneralization to Mtb unless directly tested.

Author Response

Response to Reviewer #1:

The manuscript presents a novel and well-structured investigation of the MfpA protein, linking its antibiotic resistance function to modulation of host mitochondrial dynamics and immune evasion in Mycobacterium bovis BCG infection. The study is relevant, particularly in the context of persistent tuberculosis (TB) infections and resistance development. I have two minor comments.

Response:

We thank the reviewer for their positive initial assessment of our work. We have addressed their specific concerns below.

Reviewer #1, Comment 1

Line 84–86: The authors mentioned MfpA is secreted by the bacteria and localizes to host mitochondria. However, the mechanism of secretion is not described. Consider including known or hypothesized secretion pathways with appropriate citations.

Response:

We appreciate the reviewer’s comment regarding the mechanism of MfpA secretion. We agree with the reviewer that we have not provided direct experimental evidence to conclusively confirm the secretion mechanism of MfpA in our study.

Previous studies have showed that MfpA is associated with membrane fractions, which suggests that it may be secreted through a membrane-associated transport system (Xiong et al., 2005 doi: 10.1021/pr0500049 and Målen et al., 2010 doi: 10.1186/1471-2180-10-132). 

In this study, our experiment results showed that MfpA is indeed a secreted protein, as MfpA was identified in the culture filtrate (Fig. 4A in the revised manuscript), which supports the hypothesis that it may be secreted through a membrane-associated transport system.

Additionally, within the MfpA (Rv3361c) operon, the gene Rv3365c has been previously identified as a secretion-related protein (Danelishvili L, et al., 2012 doi: 10.3389/fmicb.2011.00281), meaning that it plays a role in the secretion of other proteins or molecules from the bacterial cell. Since Rv3365c is involved in a secretion-related process, its function could be part of a system that MfpA might also be a part of.

Based on these findings and existing literature, we propose two potential secretion mechanisms for MfpA:

  1. The Sec system: This system is responsible for the secretion of a wide variety of proteins, including those that are inserted into the cell membrane. Given the membrane association of MfpA observed in our study, we hypothesize that it could be secreted via the Sec pathway, which is known for transporting proteins across the bacterial membrane.

  1. The Tat system: If MfpA is secreted in its fully folded form, the Tat system might also be involved, as this system is responsible for transporting folded proteins across the bacterial membrane.

These hypotheses have not yet been experimentally validated, and as a result, we did not include them in the manuscript. We plan to address them in future studies. Specifically, we aim to use techniques such as secretion inhibition assays and trans-membrane transport analyses to identify the precise mechanism of MfpA secretion.

To enhance the clarity and accuracy of our manuscript, we have revised the relevant section to include the following:

“Knocking out MfpA did not affect bacterial growth in vitro (Fig. 1C). The above study demonstrated that MfpA regulates the expression of TNF-α and IL-6 and affects apoptosis, suggesting that MfpA is a virulence factor. Previous studies showed that MfpA was identified in membrane fractions, which suggests that it may be secreted through a membrane-associated transport system (Xiong et al., 2005 doi: 10.1021/pr0500049 and Målen et al., 2010 doi: 10.1186/1471-2180-10-132). Additionally, within the MfpA (Rv3361c) operon, Rv3365c has been identified as a se-cretion-related protein (Danelishvili, L et al., 2011 doi:10.3389/fmicb.2011.00281), further suggesting that MfpA (Rv3361c) could be a secret-ed protein influencing host cells.”

We hope this revision better addresses the reviewer’s concern regarding the secretion mechanism of MfpA and clarifies the potential secretion pathways we are exploring. We did not include these mechanisms in the current manuscript because direct experimental evidence is not yet available. Our future experiments will aim to provide the necessary data to confirm or refine these hypotheses, and we plan to incorporate the findings into future publications.

Reviewer #1, Comment 2:

"Promotes Mtb survival by modulating the NF-κB pathway." However, the data shows this conclusion is based on M. bovis BCG. Please avoid overgeneralization to Mtb unless directly tested.

Response

We appreciate the reviewer's careful suggestions regarding the conclusions in our study. In response to your feedback, we have revised certain statements in the Abstract to avoid overgeneralizing the applicability of the BCG experimental results to M. tuberculosis (Mtb).

In the revised Abstract, we added the following phrase: " Studies using BCG provide valuable insight into Mtb’s survival mechanisms, high-lighting MfpA’s dual role in resistance and pathogenesis." This statement clarifies the potential relevance of the findings while indicating that additional research is necessary to verify their direct applicability to Mtb. We have also revised the Discussion to further clarify our findings and strengthen the overall narrative.

We believe that this modification effectively addresses the reviewer's concerns about over-generalization, while still retaining the insightfulness of our findings for Mtb biology. We greatly appreciate the reviewer’s valuable comments and hope that these revisions bring the manuscript in line with the journal's requirements.

Once again, thank you for your detailed review and constructive feedback.

Reviewer 2 Report

Comments and Suggestions for Authors

General Comments:

The study of Wang and co-authors deals with the roles of the antibiotic-resistant protein MfpA, the fluoroquinolone resistance protein A, in the pathogen Mycobacterium tuberculosis. For this purpose, authors used as a model species the species Mycobacterium bovis BCG (Bacillus Calmette-Guérin). The study presented here has a high number of interesting aspects such as a high number of experimental data and the combination of parameters using a high number of techniques. Also, potential therapeutical aspects of this research can be considered. However, important fails were detected by this Reviewer, and I consider that authors need to correct or explain in the text. Overall, (1) authors omitted the use of M. bovis BCG in important parts of the article, including abstract or keywords, and focussed in M. tuberculosis seeming that they work with this pathogen. Authors need to explain clearly without doubt that article and, consequently, results are from M. bovis BCG, not M. tuberculosis; (2) Introduction must be ameliorated and this section must finish with clear objectives; (3) basic information is missing in material and methods (brief description or references of methods such as in statistical analyses). Moreover, all reagents, costumers and equipment must be described and expressed in the same format (city, (state), country). Acronyms such as SEM must be described; (4) All methods need to be detailed with size number; (5) English language must be revised; (6) Experimental design must be enhanced: high resolution images including one or two electron microscopy techniques must be useful to validate results such as the data of apoptosis, only evaluated with a marker of early apoptosis; and, finally, (6) Conclusions must be revised and improved to be real conclusions and matching with objectives.

In my opinion, the manuscript presents a high amount of data, several of them previously described by other authors, but the text needs a major revision to increase its quality. Then, I would think it not adequate for publication in the present form in a top journal such as Cells.

Specific Comments:

-Abstract: add clearly the study was made with Mycobacterium bovis BCG, in the present form seems that authors work with Mycobacterium tuberculosis

-Introduction must include all aspects of the study (for example: apoptosis appears in title but does not appear in the Introduction or in Objective of the study. Authors need to focus on the relevant aspects of the manuscript and all sections must match. Finish Introduction section with clear objectives. Now, authors finished with an species of summary.

- Methods must be clear with adequate steps and data to repeat the experiments. Missing information. Detail methods or add adequate references with detailed the same methodologies. Size groups? Total number of cells counted? The Annexin V assay offers the possibility of detecting early phases of apoptosis before the loss of cell membrane integrity. An evaluation with electron microscopy techniques (including quantitative or semiquantitative approaches) is mandatory to relate biochemical or genetical data with real morphometric effects.

Also, statistical analysis must be enhanced (how authors checked conditions for the use of parametric tests? How authors corrected error type I of sequential analyses?) Describe acronyms as SEM.

Author Response

Response to Reviewer #2:

General Comments:

The study of Wang and co-authors deals with the roles of the antibiotic-resistant protein MfpA, the fluoroquinolone resistance protein A, in the pathogen Mycobacterium tuberculosis. For this purpose, authors used as a model species the species Mycobacterium bovis BCG (Bacillus Calmette-Guérin). The study presented here has a high number of interesting aspects such as a high number of experimental data and the combination of parameters using a high number of techniques. Also, potential therapeutical aspects of this research can be considered. However, important fails were detected by this Reviewer, and I consider that authors need to correct or explain in the text.

Response:

We greatly appreciate the reviewer’s thoughtful feedback and constructive suggestions. We understand the importance of clarity, accuracy, and the need to provide detailed information for reproducibility. We will revise the manuscript accordingly to address the concerns raised.

Reviewer #2, Comment 1:

authors omitted the use of M. bovis BCG in important parts of the article, including abstract or keywords, and focused in M. tuberculosis seeming that they work with this pathogen. Authors need to explain clearly without doubt that article and, consequently, results are from M. bovis BCG, not M. tuberculosis;

Response: 

Mycobacterium bovis BCG vs. M. tuberculosis Clarification:

We thank the reviewer for pointing out the need for clear differentiation between M. bovis BCG and M. tuberculosis. We have ensured that this distinction is explicitly made throughout the manuscript, including the Abstract, keywords, and throughout the main body of the text. We have acknowledged that the research presented in this study is based on M. bovis BCG, and we will revise the manuscript to emphasize this point to avoid any confusion. Additionally, we have clarified in the Discussion section that while M. bovis BCG serves as a model, the findings may have implications for M. tuberculosis and other pathogens, but further studies will be required to confirm this.

Reviewer #2, Comment 2:

Introduction must be ameliorated and this section must finish with clear objectives;

Response:

Introduction Section Improvement:

We appreciate the reviewer’s suggestion to enhance the Introduction. We have revised the Introduction to ensure that all aspects of the study are included, such as the role of apoptosis, which is mentioned in the title but not adequately addressed in the introduction. We have also revised the objectives to clearly reflect the study’s goals, ensuring a logical flow and aligning the Introduction with the conclusions drawn from the results. A more precise and detailed focus on the research questions have been provided, concluding with a clear statement of the study's objectives.

Reviewer #2, Comment 3:

Basic information is missing in material and methods (brief description or references of methods such as in statistical analyses). Moreover, all reagents, costumers and equipment must be described and expressed in the same format (city, (state), country). Acronyms such as SEM must be described; All methods need to be detailed with size number;

Response:

Methods Section Clarification:

We agree that the Methods section requires more clarity and detail to allow for reproducibility. We have provided a more detailed description of the statistical analyses, including how conditions for parametric tests were checked and how potential type I errors were managed in sequential analyses. We have also ensured that all reagents, equipment, and customer references are properly described, with full citation details, including city, state, and country. Furthermore, we have expanded on the experimental procedures to clarify the sample sizes, number of cells counted, and any necessary steps to repeat the experiments.

Reviewer #2, Comment 4:

English language must be revised;

Response:

English Language and Grammar:

We appreciate the reviewer’s note on the language and grammar. We have worked with MDPI’s Author Services to ensure that the manuscript is polished, with improved clarity and fluency throughout the text.

Reviewer #2, Comment 5:

Experimental design must be enhanced: high resolution images including one or two electron microscopy techniques must be useful to validate results such as the data of apoptosis, only evaluated with a marker of early apoptosis; and, finally,

Response:

Experimental Design and Image Quality:

We acknowledge the importance of high-resolution images to validate our findings, particularly with respect to apoptosis markers. We have enhanced the experimental design by including high-resolution images, including electron microscopy (EM) techniques, as suggested. This has helped to validation the results obtained from the Annexin V assay, which has identified early apoptosis. We have provided EM images in Fig.3C of the revised manuscript to reinforce the biochemical and genetic data with direct morphological evidence.

Reviewer #2, Comment 6:

Conclusions must be revised and improved to be real conclusions and matching with objectives.

Response:

Conclusion Section Improvement:

We have revised the conclusion section to more clearly align with the objectives of the study. The conclusions have been rewritten to reflect the actual findings of the study, ensuring that they are robust and directly linked to the results obtained. This revision has ensured that the conclusions are logically derived from the data and provided clear insights into the therapeutic potential of targeting mitochondrial dynamics in mycobacterial infections.

Specific Comments:

-Abstract: add clearly the study was made with Mycobacterium bovis BCG, in the present form seems that authors work with Mycobacterium tuberculosis

Response:

Abstract – Clarification of M. bovis BCG:

We appreciate the reviewer’s concern regarding the ambiguity in the Abstract, which may suggest that the study was conducted with M. tuberculosis rather than M. bovis BCG. We have revised the Abstract and added the following phrase: “Studies using BCG provide valuable insight into Mtb’s survival mechanisms, high-lighting MfpA’s dual role in resistance and pathogenesis.”

-Introduction must include all aspects of the study (for example: apoptosis appears in title but does not appear in the Introduction or in Objective of the study. Authors need to focus on the relevant aspects of the manuscript and all sections must match. Finish Introduction section with clear objectives. Now, authors finished with a species of summary.

Response:

Apoptosis and Objectives:

We agree with the reviewer that the Introduction should be more closely aligned with the manuscript’s content. In the revised version, we have explicitly introduced the role of apoptosis, and the study’s objectives are clearly stated at the end of the Introduction. We have made sure that the Introduction now provides a clear and concise overview of the research questions, with a particular focus on mitochondrial dynamics, resistance proteins, and apoptosis in M. bovis BCG.

-Methods must be clear with adequate steps and data to repeat the experiments. Missing information. Detail methods or add adequate references with detailed the same methodologies. Size groups? Total number of cells counted? The Annexin V assay offers the possibility of detecting early phases of apoptosis before the loss of cell membrane integrity. An evaluation with electron microscopy techniques (including quantitative or semiquantitative approaches) is mandatory to relate biochemical or genetical data with real morphometric effects.

Response:

Methods - Clarity and Experiment Details

We have added a more detailed description of the statistical analyses, including how parametric tests were selected and the methods used to correct for potential type I errors. We have also added additional details on the sample sizes, the number of cells counted, and clarify the procedure used for the Annexin V assay. We agree that electron microscopy techniques, including both qualitative and quantitative approaches, are necessary to support our findings regarding apoptosis. As such, we have incorporated EM images and revised the Methods to include specific experimental steps and quantitative approaches used in the analysis.

-Statistical analysis must be enhanced (how authors checked conditions for the use of parametric tests? How authors corrected error type I of sequential analyses?) Describe acronyms as SEM.

Response:

Statistical Analysis Comment

We greatly appreciate the reviewer’s valuable suggestion regarding the enhancement of the statistical analysis. To address the concern, we have revised the Methods section to include a more comprehensive description of the statistical analyses used in the study.

  1. Parametric Test Conditions:

We now explicitly mention how we assessed the normality of data distribution before applying parametric tests. For small sample sizes (n ≤ 50), we used the Shapiro–Wilk test, while for larger sample sizes (n > 50), we applied the D'Agostino–Pearson test. We also made sure to indicate that parametric tests were only applied if the data passed the normality test (P > 0.05).

  1. Type I Error Control in Sequential Analyses:

Regarding the correction for Type I errors in sequential analyses, we clarified that homogeneity of variances was tested using the Brown–Forsythe test (P > 0.05) before performing one-way ANOVA. Additionally, we performed post hoc testing (Tukey’s test) only after confirming the assumptions for parametric tests were met.

  1. Clarification of Acronyms:

As suggested, we have clarified the use of acronyms. "SEM" (Standard Error of the Mean) is now fully described in the revised manuscript.

  1. Statistical Significance:

We have also included a detailed explanation of the criteria used for determining statistical significance, indicating the P value thresholds for various levels of significance (****, P < 0.0001; ***, P < 0.001; **, P < 0.01; *, P < 0.05; ns, not significant).

We believe that these additions will address the reviewer's concerns and ensure the statistical analysis is clearly described, reproducible, and robust. Thank you for your constructive feedback, which has allowed us to improve the clarity and precision of the statistical methodology.

Round 2

Reviewer 2 Report

Comments and Suggestions for Authors

The study of Wang and co-authors deals with the roles of the antibiotic-resistant protein MfpA, the fluoroquinolone resistance protein A, in the pathogen Mycobacterium tuberculosis. For this purpose, authors used as a model species the species Mycobacterium bovis BCG (Bacillus Calmette-Guérin). The study presented here has a high number of interesting aspects such as a high number of experimental data and the combination of parameters using a high number of techniques. Also, potential therapeutical aspects of this research can be considered. However, important fails were detected by this Reviewer in the first version of the manuscript. Now, the revised version of the manuscript was greatly improved by the authors that increased information and accuracy of data through all the text. Authors also have added new experimental data with transmission electron microscopy that enhance the quality of the study and render more consistent the conclusions of this research.

In my opinion, the manuscript presents a high amount of data and techniques and both experimental design and text were clearly improved by the authors. Then, I would think it adequate for publication in the present form in a top journal such as Cells.